# Therapeutic Opportunities for Biomarkers in Metastatic Spine Tumors

**DOI:** 10.3390/cancers16183152

**Published:** 2024-09-14

**Authors:** Christian Schroeder, Beatrice Campilan, Owen P. Leary, Jonathan Arditi, Madison J. Michles, Rafael De La Garza Ramos, Oluwaseun O. Akinduro, Ziya L. Gokaslan, Margot Martinez Moreno, Patricia L. Zadnik Sullivan

**Affiliations:** Department of Neurosurgery, Warren Alpert Medical School of Brown University, Providence, RI 02903, USA

**Keywords:** spinal metastases, primary tumor, molecular therapy, molecular markers, targeted therapy

## Abstract

**Simple Summary:**

This research highlights the differences in genetic mutations between primary tumors and their subsequent spine metastases. These distinctions are relevant to neurosurgeons because effective surgical planning and goals rely on factors such as patient prognosis, which is impacted by tumor genetic markers. Certain genetic markers may be more or less favorable, thus impacting the predicted patient treatment course and prognosis. The authors aim to present an overview of these complex molecular characteristics for neurosurgical oncology providers. The content of this review is intended to assist neurosurgeons’ decision-making when caring for spinal oncology patients and to illustrate the importance of multidisciplinary management of this complex patient population.

**Abstract:**

For many spine surgeons, patients with metastatic cancer are often present in an emergent situation with rapidly progressive neurological dysfunction. Since the Patchell trial, scoring systems such as NOMS and SINS have emerged to guide the extent of surgical excision and fusion in the context of chemotherapy and radiation therapy. Yet, while multidisciplinary decision-making is the gold standard of cancer care, in the middle of the night, when a patient needs spinal surgery, the wealth of chemotherapy data, clinical trials, and other medical advances can feel overwhelming. The goal of this review is to provide an overview of the relevant molecular biomarkers and therapies driving patient survival in lung, breast, prostate, and renal cell cancer. We highlight the molecular differences between primary tumors (i.e., the patient’s original lung cancer) and the subsequent spinal metastasis. This distinction is crucial, as there are limited data investigating how metastases respond to their primary tumor’s targeted molecular therapies. Integrating information from primary and metastatic markers allows for a more comprehensive and personalized approach to cancer treatment.

## 1. Introduction

The spinal column is a particularly common site of metastatic cancer, with bony spinal metastases identified in >10% of all cancer patients [1]. Advances in radiation and chemotherapy have prolonged survival, making spinal metastases an area of focus due to the unique role of surgical treatment for metastatic disease [2,3]. It is well known that surgery plus radiation provides the greatest survival benefit for patients with metastatic spine disease [4]. Advances in molecular therapies have resulted in new drugs that can mitigate disease with fewer side effects than traditional cytotoxic chemotherapy. While surgery remains a mainstay in cancer treatment, surgical decision-making must be informed by current paradigms in medical management to optimize patient care. 

One such paradigm is the growing influence of tumor genetic and molecular markers on various facets of oncological care. Primary tumor molecular markers are essential for cancer diagnosis, disease subtyping, and selection of the most appropriate treatment approach. Additionally, these primary markers can provide information about the likely course of the disease, assess the aggressiveness of the tumor, and predict prognosis, which may subsequently guide the extent of surgery and goals of surgical reconstruction.

Recent work has begun to examine cellular and molecular mechanisms regulating metastasis from specific primary tumors, including the recent identification of distinct classes of stem cells that may contribute to vertebral body engraftment [5]. While this field remains in its infancy, further insights may lead to new molecular targets and ultimately reduce or refine the indications for surgery, which carries risks despite offering quality-of-life improvements [1,6,7,8,9,10].

The molecular profiles of metastatic lesions may greatly differ from those of the primary tumor but may also be related in important ways. It has been suggested that the nature of the primary seeding cancer cell determines subsequent metastatic properties with respect to growth and response to therapy [2]. Identifying molecular differences between primary and metastatic tumors is crucial for selecting effective treatment options, determining therapeutic resistance, and monitoring the tumor’s response, as changes in the molecular profile of metastatic lesions can indicate how a cancer is responding to treatment [2,11]. 

Epigenetic changes are also drivers of cancer tumorigenesis and, therefore, may constitute a unique primary and metastatic differential marker [12,13]. Cancer cells exhibit a unique epigenetic landscape depending on associated immune cells and other components of the tumor microenvironment (TME). 

The goal of this paper is to describe and compare the molecular profiles of primary and metastatic tumors in the spine. In this paper, we focus on the four most common primary lesions to metastasize to the spine—lung, breast, prostate, and renal—although gastrointestinal tumors, such as pancreatic ductal adenocarcinoma, and thyroid cancer, such as follicular thyroid carcinoma, can metastasize to the spine in rare cases [14]. This review is useful to the practicing neurosurgeon because it offers the opportunity for education on the multidisciplinary management of spinal oncology patients and the impacts of such management on surgery, which has direct and treatment-altering implications for surgeons and patients alike. 

## 2. Lung Cancer

Lung cancer has a global incidence of 2 million people, with 1.7 million people succumbing to the disease each year [15]. Lung cancer is classified into small cell lung cancer (SCLC) and non-small cell lung cancer (NSCLC), although rarer types of lung cancer, such as carcinoid tumors, do exist. NSCLC can then be further differentiated into adenocarcinoma, large cell carcinoma, and squamous cell carcinoma. Due to their anatomical location, primary lung tumors may directly invade the spine, such as in cases of the superior sulcus or Pancoast lesions, which may present with involvement of the cervicothoracic junction [16].

### 2.1. Primary: SCLC

Although the incidence of SCLC is declining, potentially due to a decrease in smoking, outcomes for patients with SCLC remain poor [17,18]. The high incidence of this disease and its metastatic tropism to bone, which occurs in 20% of patients, make SCLC a common concern for spine surgeons [19]. Radiation with platinum-based chemotherapy is common practice for the treatment of SCLC; however, surgery may play a role in cases of spinal cord compression or instability. Many mutations have been discovered in primary SCLC, which include mutations in the retinoblastoma gene (RB), the rge myc proto-oncogene family, the phosphatase and tensin homolog gene (PTEN), and the fibroblast growth factor 1 receptor gene (FGFR1) [20,21,22,23]. Despite the discovery of these mutations in SCLC, there remain very few targeted therapies. Current therapies for primary SCLC include programmed cell death ligand 1 (PD-L1) inhibitors as the first line treatment, although anti-cytotoxic T-lymphocyte associated protein 4 (CTLA-4) antibodies have also been shown to be effective against the disease [24,25,26].

### 2.2. Spinal Metastasis: SCLC

In lung cancer metastatic to bone, FGFR1 is often mutated, and it is presumed that overexpression of this gene increases tumorigenesis in the bone [27]. This is thought to be a result of significant expression of fibroblast growth factors (FGFs) in bone, thus making FGFR1 upregulation a favorable mutation for bony invasion [27]. Overexpression of nuclear factor I B (NFIB), a previously reported oncogene in mice, has also demonstrated increased bone metastasis rates and offers a potential target for bone-specific spine metastasis [28]. In particular, high levels of NFIB have been associated with increased dedifferentiation and increased invasiveness [28]. Beta3-Integrin, a receptor for many bone matrix proteins, has been shown to be overexpressed, particularly in SCLC metastatic to bone, as it plays a role in tumor cell migration, survival, and proliferation [29]. Additionally, C-X-C motif chemokine receptor 4 (CXCR4), a G-protein-coupled receptor, tends to be overexpressed in certain cell types, which are particularly associated with bone metastases [30]. 

These markers may have useful implications for prognosis and subsequent surgical decision-making. FGFR1-amplified SCLC has been shown to only partially respond to FGFR1 inhibitors, therefore suggesting that the presence of this marker may be associated with a worse prognosis [31]. Similarly, NFIB overexpression seems to be an indication of increased tumor aggressiveness on a molecular level, and Beta3-integrin has been noted to play a central role in resistance to epidermal growth factor receptor tyrosine kinase inhibitors (EGFR-TKIs) [32]. CXCR4 remains understudied in solid tumors with regard to drug targeting, and upregulation of CXCR4 has been implicated in the development of drug-resistant lesions following chemotherapy. Thus, it is a likely marker of a worse prognosis [33]. Overall, any of the above molecular markers in a sample of SCLC metastatic to the spine could be an indication to consider reformulating the surgical plan. In general, spinal metastasis surgery is not recommended for any patient with a prognosis shorter than 3 months, and due to the poor prognosis of SCLC metastatic to the spine at baseline, surgery may not be a viable option for some patients harboring lesions with the above molecular markers. 

### 2.3. Primary: NSCLC

The most common mutations in primary NSCLC are epidermal growth factor receptor 1 (EGFR), ERBB2 (also known as HER2 and NEU), and KRAS [34]. EGFR, MET, and RET mutations, in particular, are more common in smokers than non-smokers [34]. Lung adenocarcinoma is strongly associated with smoking history and is included within the NSCLC group [35]. With EGFR being one of the most common mutations, the creation of EGFR-TKIs was transformative for NSCLC treatment and has had significant impacts on NSCLC prognosis. Other potential therapeutic targets include the PI3K/Akt/mTOR pathway, which is significant in the development of NSCLC and is being explored by multiple clinical trials [25,36]. However, most of these trials are being applied to NSCLC patients without molecular testing. ALK inhibitors are another class of drugs commonly used for NSCLC patients. ALK mutations are uniquely more common in non-smokers than smokers and are seen in about 15% of non-smoking NSCLC patients [37]. For patients with BRAF mutations, BRAF inhibitors have shown some success in both primary and metastatic tumors in one study, although the metastatic tumors in this study were not specific to bone or the spine [38].

### 2.4. Spinal Metastasis: NSCLC

Data on metastatic NSCLC to bone are very limited. Conflicting preliminary data state that patients with EGFR and HER2 mutations may be at a higher risk for developing metastases to bone [39]. In one study, patients treated with TKIs who had EGFR-mutated NSCLC to the spine showed increased survival but no improvements in pain, ambulation, or neurologic status [40]. Importantly, in this study, patients undergoing surgery for NSCLC metastatic to the spine who received EGFR-TKIs tended to live longer than those receiving traditional cytotoxic chemotherapy [40]. This difference was not shown to be statistically significant, but the authors theorized that this could be due to a lack of genetic testing of metastatic tumors. Genetic testing of metastatic tumors is essential because NSCLC demonstrates inconsistent EGFR mutations between primary and metastatic lesions [41]. If EGFR-mutated NSCLC spinal metastases are shown to respond to EGFR-TKIs, patients with such lesions might be more routine surgical candidates due to their increased survival and subsequent increased need for pain palliation via surgery. This underscores the need for a biopsy of the metastatic lesion to guide subsequent medical and surgical treatment. 

## 3. Breast Cancer

With a global incidence of over 2.26 million people in 2020, breast cancer is the most common malignant tumor threatening women’s health [42,43]. In the United States alone, there are more than 3.8 million women living with a history of breast cancer, within which considerable heterogeneity in demographic, pathological, and prognostic characteristics exist [44,45]. Although overall mortality rates have declined in the past few decades due to the advancement of diagnostic and therapeutic methods, it has remained one of the most prevalent cancers worldwide [46,47].

### 3.1. Primary

Key molecular biomarkers such as estrogen receptor (ER), progesterone receptor (PR), and human epidermal growth factor receptor 2 (HER2) drive treatment decisions in primary breast cancer [48], and mutations in genes like ESR1, PIK3CA, BRCA1, and BRCA2 inform personalized therapies [49,50,51,52]. For triple-negative breast cancer (TNBC), Syndecan-1 (CD138) and PD-L1 hold promise as prognostic markers [53,54].

Serum biomarkers are an emerging avenue in the characterization and management of breast cancer [55]. CEA, CA153, and CA125 are among the most commonly cited and have been closely associated with aggressive clinicopathological features [56]. Both the American Society of Clinical Oncology (ASCO) and the National Comprehensive Cancer Network (NCCN) have put forth guidelines that advocate for the continued monitoring of CEA, CA153, and CA27-29 in breast cancer patients [55,56]. Tissue polypeptide-specific antigen (TPS) has also been noted for its elevated levels and potential role in predicting poor outcomes, though it has yet to be established in clinical practice [47]. Optimal diagnostic accuracy and efficiency may be achieved through parallel testing of serum biomarkers or even their use in conjunction with routine imaging [57].

Continued investigation into the underlying mechanisms of breast cancer has unveiled a variety of potential targets and novel treatments: mammalian target of rapamycin (mTOR); NTRK1, NTRK2, NTRK3; Cyclin D, Cyclin E; CDK4/6; B-Myb; Twist; DMP1β; TP53; FGFR, AKT1, AKT2, Src, PTEN, KRAS, APC, NF1, MAP2K4, MAP3K1, uPA, and PAI-1 [56,58,59,60,61,62,63,64,65,66].

### 3.2. Spinal Metastasis

Spinal metastases represent a significant and often severe consequence of breast cancer. Breast cancer has a particular affinity for the spine: although only 8% of all patients with breast cancer developed metastases to bone, about two-thirds of all reported osseous metastases were to the spine [67,68]. Of these lesions to the spine, an estimated one-third become symptomatic with deterioration in quality of life, intractable pain, neurological deficits, and mechanical instability [69]. The severity and symptomology of breast cancer metastatic to the spine have historically driven treatment to be primarily palliative [70]. Available options aimed toward pain relief and the restoration and preservation of neurological function include pharmacologic management, radiotherapy, and surgery [71]. 

Considerably less information is available regarding molecular markers for spine metastases. Rather, assessment for therapeutic approaches tends to rely on the genotypic subtype of the primary cancer. As characterization of ER, PR, and HER2 status is the gold standard of breast cancer treatments, much of the existing literature tends to focus on drawing associations between these markers, metastatic patterns, and prognostic assessment.

Although limited, studies show differences in survival following spinal metastases between different genomic subtypes [71]. In a French national database, patients with the HER2+ subtype exhibited a median overall survival of 76.1 months compared to 17.3 months in the TNBC group [71]. Consequently, the authors argued for the determination of the breast cancer genomic subtype as a key prognostic factor for patients with spinal metastases. Other studies have elucidated how molecular tumor typing can shed light on clinical outcomes, such as the 60-month mean survival time in luminal phenotypes compared to the 15–19 months seen in TNBC cases [72].

The incidence of site-specific bone metastases from breast cancer and the underlying hormonal receptor subtypes has also been explored [73,74]. CX3C motif chemokine receptor 1 (CX3CL1), for example, has been reported to play an anti-tumor role in spinal metastases, but more recent studies have postulated that CX3CL1 expression is associated with a worse prognosis [75,76]. While more research is needed to precisely elicit the role of CX3CL1 in breast cancer metastatic to the spine, it could serve as a molecular indication of prognosis in the future.

The ongoing debate concerning the precise functions of these molecular markers in spinal metastases of breast cancer underscores the vast potential of molecular tumor typing with regard to treatment goals and prognostication. The lack of exploration into the molecular markers in spinal metastatic breast cancer lesions reveals a significant limitation in surgeons’ and oncologists’ abilities to effectively predict disease progression and subsequent treatment plans. The global incidence of spinal metastases in breast cancer patients supports the urgent need for further study of molecular markers and actionable mutations to ensure that these patients are receiving neurosurgical care based on accurate prognoses and potential for benefit.

## 4. Prostate Cancer

Following lung cancer, prostate cancer is the most common cause of cancer in adult men, with metastatic castration-resistant prostate cancer remaining a leading cause of death in this population despite increased screening protocols and the development of androgen receptor antagonists [60,77,78,79]. Prostate metastases have a predilection for the spine, and the median overall survival after diagnosis of metastases is estimated to be about 2 years [78,79].

### 4.1. Primary

Historically, androgen deprivation therapy (ADT) for hormone-sensitive prostate cancer has comprised the mainstay of systemic therapy in this primary prostate cancer, often in combination with androgen receptor signaling inhibition (ARSI) with abiraterone, enzalutamide, or apalutamide, with or without docetaxel [80]. In general, androgen receptor antagonists have offered improved outcomes in this disease. Treatment for castration-resistant prostate cancer (CRPC), however, has been more challenging. Non-targeted initial treatment regimens often include abiraterone with prednisone, apalutamide, darolutamide, or enzalutamide [81].

Molecular therapies have recently been employed in specific subsets of CRPC patients, particularly those with BRCA mutations and prostate-specific membrane antigen (PSMA) positivity. BRCA-mutated prostate cancers are susceptible to Poly ADP-Ribose Polymerase (PARP) inhibitors, such as Olaparib, to the extent that BRCA testing has become routine [7,82]. Separately, cases harboring prostate-specific membrane antigen (PSMA) positivity as detected via PSMA PET scan can be treated with PSMA-targeted radioligand therapy, though this has not shown long-term improvement in overall survival compared with cabazitaxel [77,83,84]. PI3 kinase/AKT signaling-related mutations, such as the PTEN mutation, are associated with worse outcomes in CRPC. These cases can be treated with ipatasertib, an inhibitor of AKT, which has shown modest benefit [84].

Among other targets and therapeutic strategies tested, molecular strategies targeting PD-1 (pembrolizumab), MET/VEGFR2 (cabozantinib), EGFR (cetuximab), TKIs (gefitinib, erlotinib, or lapatinib), and CTLA4 (ipilimumab) have demonstrated little improvement in clinical endpoints in trials [79,85,86,87,88,89,90]. The active cell-based autologous immunotherapy sipuleucel-T, consisting of peripheral mononuclear cells activated with a prostate antigen fusion protein, resulted in modest improvements in overall survival in clinical trials, but it was discontinued commercially after FDA approval and is not available as a treatment regimen [91]. Bispecific T-cell therapies, which may offer more specific targeting of molecular therapies to metastatic CRPC cells, have recently been investigated as well, and these represent a promising future investigation and area of active study. No large trial results studying overall or progression-free survival with these approaches are yet available [77,89].

### 4.2. Spinal Metastasis

Few biomarkers that specifically differentiate prostate bone (or vertebral) metastases from primary lesions have been definitively recognized, though mechanisms that precipitate metastasis to the bone have been hypothesized based on recent data [92,93,94,95,96]. Relative to patients with localized disease, patients with metastatic prostate cancer have been noted to have a higher prevalence of germline mutations in multiple genes involved in DNA repair processes, including BRCA2, ATM, CHEK2, BRCA1, rad51d, and PALB2, which are typically associated with more aggressive disease [93,97]. A recent study that conducted microarray-based computational analysis of available datasets identified six genes associated with prostate bone metastasis samples (DDX47, PRL17, AS3MT, KLRK1, ISLR, and S100A8) and further identified the prognostic significance of several of these for survival prediction [95]. Specifically, DDX47, RPL17, AS3MT, KLRK1, and S100A8 were found to be markers of worse prognosis in bony metastases. Zhuang et al. examined genes related to “stemness,” a measure of how closely tumor cells resemble cancer stem cells, which differentiate primary versus metastatic disease, and identified that known transcription factor Forkhead Box M1 (FOXM1) and associated DNA glycosylase NEIL3 may contribute to metastasis and disease progression, building on past work identifying a miRNA-regulated metastasis pathway also involving FOXM1 [94,95,96,97,98]. This pathway is of particular interest because of its role in the development of resistance to enzalutamide, a targeted anti-androgen treatment [98]. Therefore, biomarkers associated with these pathways may be associated with a worse prognosis and subsequent changes to medical and surgical management.

## 5. Renal Cell Cancer

Renal cell carcinoma (RCC) is one of the top ten diagnosed malignancies in both men and women, accounting for 5% of all malignancies in males and 3% in females [99]. RCC is characterized by its complex classification schema, high metastatic tendency, and resistance to conventional radiation and chemotherapy [100,101]. Radical or partial nephrectomy remains the most effective therapy for primary disease, but even this treatment presents challenges. 

### 5.1. Primary

Diagnosis of RCC is generally based on morphological and immunohistochemical properties of tissue samples, though chromosome 3p alteration and especially VHL gene mutational status are the molecular pathology features recognized as strongest support for clear cell RCC diagnosis, and evolving profiles of markers exist for the other subtypes as well [102]. Polybromo 1 (PBRM1) gene mutations are also highly specific to clear cell RCC [102]. RCC is increasingly viewed as a fundamentally metabolic disease, with mutational changes across many genes contributing to metabolic dysregulation [103]. A paradigm shift towards targeted therapies, including TKIs, mTOR inhibitors, and monoclonal antibodies against VEGF, has resulted in prolonged overall and progression-free survival in metastatic disease [104,105]. Despite these advances, individual patient responsiveness to antiangiogenic therapies and other treatments is notoriously variable and unpredictable in RCC [106]. Accordingly, patient-specific molecular markers to enable a more personalized treatment approach have become an area of recent focus.

### 5.2. Metastasis

RCC metastases to the spine share the primary disease’s resistance to conventional therapy but are notably strong candidates for surgical spondylectomy since most cases consist of a singular spinal lesion [100,107]. Because of this dire prognostic landscape and the existing dearth of established RCC biomarkers in any application, new molecular insight is needed to identify new molecular markers as prognostic, discriminative, and therapeutic targets. 

Although VHL gene inactivation and carbonic anhydrase IX (CAIX) are established biomarkers for RCC, the prognostic utility of these molecular markers remains unclear [106,108]. Accordingly, epigenetic differences between metastatic RCC tissue (including specifically spinal metastases) and primary RCC tumors have not been characterized. There is some indication that microbiome differences may also comprise detectable markers of metastatic disease; however, these studies were also not specific to osseous metastases [109,110,111]. Taken together, insufficient information exists to effectively predict the clinical benefit a patient will garner from any particular molecular therapy, which makes prognostication inherently difficult. Additional study of tissue obtained from spinal metastases, given the ready access to these tissues through surgery as the mainstay of treatment for solitary metastatic lesions, may enable future advances in identifying novel targets in advanced disease as well as biomarkers for prognosis. Clinical trials targeting biomarkers discussed in this review are summarized in Table 1. 

## 6. Discussion

When cancer metastasizes to the spine, it creates a complicated clinical scenario requiring multidisciplinary management. The patient prognosis, response to medical management, and predicted benefit of surgery are all variables that are considered in the choice to pursue surgical intervention, a decision that can be informed by various prognostication systems. The NOMS framework has been well established to guide surgical decision-making by considering the neurological impact of the tumor, the specific susceptibility to medical and radiological treatments of the tumor, mechanical stability, and the overall systemic impact of treatment on the patient [112]. The Tokuhashi scoring system aims to guide surgical decision-making in metastatic spine tumors by considering the type of primary lesion, the extent of metastatic bone lesions, both vertebral and extra-vertebral, the extent of visceral metastases, and spinal cord function [113,114]. The Tomita scoring system includes consideration of primary tumor type, presence of visceral metastasis, presence of bone metastases, and the number of metastases present [113,114]. The NOMS framework, in particular, is uniquely poised to accommodate new developments in oncology as it emphasizes the interdisciplinary management of neurosurgical oncology patients [114]. This suggests that the NOMS framework may be most useful in the context of molecular profiling of metastatic lesions, as it allows for consideration of non-surgical management and the projected response to therapy. A metastatic lesion’s predicted response to therapy is often estimated according to that of the primary lesion in practice. However, as we have demonstrated, the molecular signature of the primary tumor may not remain the same when the tumor metastasizes. The differences between primary and metastatic lesions in terms of these biomarkers are summarized in Figure 1. While recent studies have suggested a high (>96%) level of genetic marker concordance between primary and metastatic lesions of cancers that commonly metastasize to the spine, differences in epigenetic profiles may be a source of further investigation. Further, while a majority of the genomes were concordant in previous studies, there were notable differences in clinically actionable mutations within the samples [115]. The bone microenvironment is quite distinct from that of primary tumor tissues in metastatic disease, consisting of a variety of cell types with which cancer cells must necessarily interact [116]. Accordingly, analysis of osseous metastatic tissue might support novel target discovery. This makes the practical application of the NOMS framework more challenging. The use of the NOMS framework and other scoring systems is meant to aid surgeons in determining whether spine surgery will be beneficial in a particular patient, but cancer remains a challenging disease to prognosticate. Further, the existence of various scoring systems underscores the fact that no one system is ideal, and in many centers, a scoring system or formal decision-making framework may not be used to direct a patient’s care [117,118]. Alongside the use of these decision-making systems, prioritization must be given to the identification of clinically actionable and prognosis-impacting mutations in metastatic lesions, as these genetic markers can have direct impacts on patient care and surgical planning. Intraoperative biopsy at the time of spinal metastasis surgery, as well as discussion with medical oncology and radiation oncology, can guide treatment plans and clarify clinically actionable mutations within the metastatic lesion.

These findings further highlight the need for biopsy at the time of metastatic tumor surgery, as well as communication with the patient’s primary oncology team. Send-out molecular panels are highly variable between institutions and may be tailored to an institution’s clinical trials or research interests. Sending a sample of the patient’s metastasis may qualify them for a new therapy.

## 7. Conclusions

Understanding the correlation between the primary and their metastatic molecular markers may serve to predict metastatic sites, find potential biomarkers that may be screened in blood biopsies, and predict prognosis and therapy response [119,120]. Therefore, there exists a significant need to identify metastatic spine molecular markers in order to determine optimal oncological and neurosurgical treatment. Integrating information from both primary and metastatic markers allows for a more comprehensive and personalized approach to cancer treatment.

## Figures and Tables

**Figure 1 cancers-16-03152-f001:**
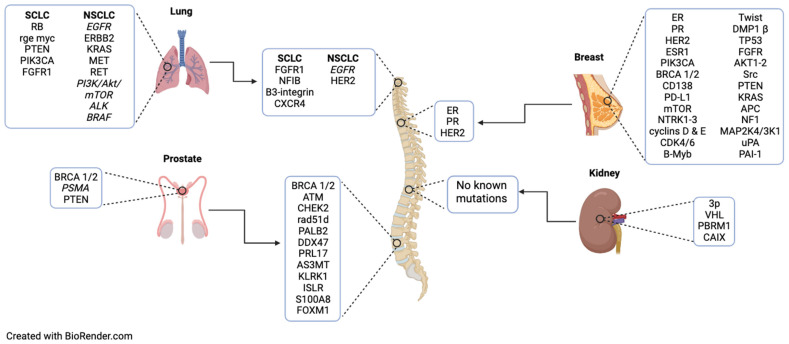
Genetic mutations in primary lung, breast, prostate, and renal cancers and their subsequent mutations upon metastasis to the spine. Mutations in italics represent known therapeutic targets discussed in this study.

**Table 1 cancers-16-03152-t001:** Relevant Clinical Trials and Biomarker Targets.

Citation	Study Type	Biomarker/ Target	Primary Lesion Type Targeted	Drug Trialed
Paz-Ares et al., 2019 [24]	Phase 3 Trial	PD-L1, CTLA-4	Lung (SCLC)	Durvalumab, tremelimumab
Horn et al., 2018 [25]	Phase 3 Trial	PD-L1	Lung (SCLC)	Atezolimab
Paz-Ares et al., 2022 [26]	Phase 3 Trial	PD-L1	Lung (NSCLC)	Nivolumab, ipilimumab
Fisher et al., 1998 [48]	Phase 1 Trial	Estrogen receptors	Breast	Tamoxifen
André et al., 2019 [51]	Phase 3 Trial	PIK3CA (in hormone receptor positive, HER2-negative patients)	Breast	Alpelisib
Robson et al., 2017 [52]	Phase 3 Trial	PARP (in BRCA mutation and HER2-negative patients)	Breast	Olaparib
Schmid et al., 2018 [54]	Phase 3 Trial	PD-L1 (in triple-negative patients)	Breast	Atezolizumab
de Bono et al., 2020 [80]	Phase 3 Trial	PARP	Prostate	Olaparib
Sartor et al., 2021 [82]	Phase 3 Trial	PSMA	Prostate	Lutetium-177-PSMA-617
Hofman et al., 2021 [83]	Phase 2 Trial	PSMA	Prostate	[177Lu]Lu-PSMA-617
Sweeney et al., 2021 [84]	Phase 3 Trial	PI3K/AKT	Prostate	Ipatasertib
Petyrlak et al., 2021 [87]	Phase 3 Trial	PD-1	Prostate	Pembrolizumab
Graff et al., 2021 [88]	Phase 3 Trial	PD-1	Prostate	Pembrolizumab
Kantoff et al., 2010 [91]	Phase 3 Trial	T-cells	Prostate	Sipuleucel-T

## Data Availability

The original contributions presented in the study are included in the article; further inquiries can be directed to the corresponding author.

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
