# Peer review of "Therapeutic Opportunities for Biomarkers in Metastatic Spine Tumors"

_cancers, 2024, doi:10.3390/cancers16183152_

Round 1
Reviewer 1 Report
Comments and Suggestions for Authors
The manuscript titled,” Therapeutic Opportunities for Biomarkers in Metastatic Spine Tumors” is very interesting and well written overall. I have a few suggestions as follows:
1. Authors should include a table highlighting any clinical trials using the biomarkers mentioned in the text.
2. Authors should also discuss about the relevance of PDAC in context of this paper, and also include that in the figure.
Author Response
Comment 1: Authors should include a table highlighting any clinical trials using the biomarkers mentioned in the text.
Response 1: The authors thank the reviewer for the suggestion and have added Table 1 summarizing the clinical trials discussed in the manuscript and their relevant biomarker targets. Table 1 can be found at line 333 on page 8.
Comment 2: Authors should also discuss about the relevance of PDAC in context of this paper, and also include that in the figure.
Response 2: The authors thank the reviewer for the suggestion. We have included the possibility of pancreatic ductal adenocarcinoma metastasis to the spine in line 71 in the fifth paragraph of page 2. Pancreatic ductal adenocarcinoma metastasizes to the spine so rarely that our institution has not seen a case in nine years. While we value this suggestion and agree that it is an interesting condition to explore, we have decided that since this manuscript focuses on the four most common tumors to metastasize to the spine, we do not feel that a discussion of pancreatic ductal adenocarcinoma metastasis should be included in this particular study.
Reviewer 2 Report
Comments and Suggestions for Authors
Dear Editor,
Thank you for inviting me to review the manuscript entitled: “Therapeutic Opportunities for Biomarkers in Metastatic Spine Tumors” submitted for publication in Cancers.
The authors present an overview of relevant molecular biomarkers in patients with spinal metastases and therapies that affect patients’ survival in regards to primary tumors of the lung, breast, prostate and kidney.
Some comments below:
1) In the abstract, line 23 “when a patient needs surgery” do the authors mean spinal surgery? This has to be clarified.
2) Lines75-81, refer also to carcinoid as a rarer type of lung cancer.
3) Line 83, the authors state that the decline of SCLC is linked to decline in smoking. Please provide some extra relevant references for this statement.
4) The authors provide a thorough overview of biomarkers for each type of cancer. I would suggest to add some figures or tables highlighting the differences between the primary and metastatic lesions in terms of these biomarkers to make it more evident to surgeons their importance.
5) In the discussion, lines 327-330. This is an important part of spinal surgery in metastatic cases. In my opinion, it should be enriched. I would suggest to talk in brief about other prognostication systems besides NOMS, such as the Tokuhashi scoring system. Here are some useful relevant references that the authors may use:
Alpantaki K, Ioannidis A, Raptis K, Spartalis E, Koutserimpas C. Surgery for spinal metastatic tumors: Prognostication systems in clinical practice (Review). Mol Clin Oncol. 2020 May;12(5):399-402. doi: 10.3892/mco.2020.2008. Epub 2020 Feb 27
Pérez-Romasanta LA, Arana E, Kovacs FM, Royuela A. The Management of Metastatic Spinal Cord Compression in Routine Clinical Practice. Cancers (Basel). 2023 May 18;15(10):2821. doi: 10.3390/cancers15102821.
Focus on their usefulness in clinical practice and the decision-making in surgery.
In general, the article is educative and well-written. In my opinion, some tables/figures would add clarity in explaining the differences of importance of each biomarker in the primary and metastatic tumor. The discussion of prognostication systems should also be enhanced.
With best regards
Author Response
Comment 1: In the abstract, line 23 “when a patient needs surgery” do the authors mean spinal surgery? This has to be clarified.
Response 1: The authors thank the reviewer for this suggestion and have added the word “spinal” to the referenced sentence in line 23.
Comment 2: Lines75-81, refer also to carcinoid as a rarer type of lung cancer.
Response 2: The authors thank the reviewer for this suggestion and have added that carcinoid tumors may also occur in the lungs in lines 80-81 in paragraph 6 on page 2.
Comment 3: Line 83, the authors state that the decline of SCLC is linked to decline in smoking. Please provide some extra relevant references for this statement.
Response 3: The authors thank the reviewer for this suggestion and have added a reference to support the claim that SCLC has been linked to smoking and therefore that a decrease in smoking may be linked to a decreased incidence of SCLC. The reference is as follows:
- Chen J, Qi Y, Wampfler JA, et al. Effect of cigarette smoking on quality of life in small cell lung cancer patients. Eur J Cancer. 2012;48(11):1593-1601. doi:10.1016/j.ejca.2011.12.002.
Comment 4: The authors provide a thorough overview of biomarkers for each type of cancer. I would suggest to add some figures or tables highlighting the differences between the primary and metastatic lesions in terms of these biomarkers to make it more evident to surgeons their importance.
Response 4: The authors thank the reviewer for this suggestion. We feel that Figure 1 summarizes the relevant biomarkers for primary and metastatic lesions but agree that the verbiage describing the figure could be clearer. Therefore, we have added the description, “The differences between primary and metastatic lesions in terms of these biomarkers is summarized in Figure 1” in lines 356-357 in paragraph 1 on page 9.
Comment 5: In the discussion, lines 327-330. This is an important part of spinal surgery in metastatic cases. In my opinion, it should be enriched. I would suggest to talk in brief about other prognostication systems besides NOMS, such as the Tokuhashi scoring system. Here are some useful relevant references that the authors may use:
Alpantaki K, Ioannidis A, Raptis K, Spartalis E, Koutserimpas C. Surgery for spinal metastatic tumors: Prognostication systems in clinical practice (Review). Mol Clin Oncol. 2020 May;12(5):399-402. doi: 10.3892/mco.2020.2008. Epub 2020 Feb 27
Pérez-Romasanta LA, Arana E, Kovacs FM, Royuela A. The Management of Metastatic Spinal Cord Compression in Routine Clinical Practice. Cancers (Basel). 2023 May 18;15(10):2821. doi: 10.3390/cancers15102821.
Focus on their usefulness in clinical practice and the decision-making in surgery.
Response 5: The authors thank the reviewer for this suggestion. We have expanded the discussion to include the following:
“The patient prognosis, response to medical management, and predicted benefit of surgery are all variables that are considered in the choice to pursue surgical intervention, a decision that can be informed by various prognostication systems.” (lines 337-340, paragraph 1, page 9)
“The Tokuhashi scoring system aims to guide surgical decision-making in metastatic spine tumors by considering the type of primary lesion, the extent of metastatic bone lesions, both vertebral and extra-vertebral, the extent of visceral metastases, and spinal cord function [114, 115]. The Tomita scoring system includes consideration of primary tumor type, presence of visceral metastasis, presence of bone metastases, and the number of metastases present [114,115]. The NOMS framework in particular is uniquely poised to accommodate new developments in oncology as it emphasizes interdisciplinary management of neurosurgical oncology patients [115]. This suggests that the NOMS framework may be most useful in the context of molecular profiling of metastatic lesions as it allows for consideration of non-surgical management and the projected response to therapy. A metastatic lesion’s predicted response to therapy is often estimated according to that of the primary lesion in practice.” (lines 343-354, paragraph 1, page 9)
“The use of the NOMS framework and other scoring systems is meant to aid surgeons in determining whether spine surgery will be beneficial in a particular patient, but cancer remains a challenging disease to prognosticate. Further, the existence of various scoring systems underscores the fact that no one system is ideal, and in many centers, a scoring system or formal decision-making framework may not be used to direct a patient’s care [118, 119]. Alongside use of these decision-making systems, prioritization must be given to identification of clinically actionable and prognosis-impacting mutations in metastatic lesions, as these genetic markers can have direct impacts on patient care and surgical planning.” (lines 366-375, paragraph 1, pages 9-10)
The authors thank the reviewer for the suggested references and have included them as references #115 and 118.
Round 2
Reviewer 2 Report
Comments and Suggestions for Authors
Dear Authors,
I appreciate the edits. No further comments.
With best regards